# AC-73 and Syrosingopine Inhibit SARS-CoV-2 Entry into Megakaryocytes by Targeting CD147 and MCT4

**DOI:** 10.3390/v16010082

**Published:** 2024-01-04

**Authors:** Isabella Spinello, Ernestina Saulle, Maria Teresa Quaranta, Elvira Pelosi, Germana Castelli, Annamaria Cerio, Luca Pasquini, Ornella Morsilli, Maria Luisa Dupuis, Catherine Labbaye

**Affiliations:** 1National Center for Drug Research and Evaluation, Istituto Superiore di Sanità, 00161 Rome, Italy; isabella.spinello@iss.it (I.S.); ernestina.saulle@iss.it (E.S.); mariateresa.quaranta@iss.it (M.T.Q.); marialuisa.dupuis@iss.it (M.L.D.); 2Department of Oncology and Molecular Medicine, Istituto Superiore di Sanità, 00161 Rome, Italy; elvira.pelosi@iss.it (E.P.); germana.castelli@iss.it (G.C.); annamaria.cerio@iss.it (A.C.); 3Core Facilities, Istituto Superiore di Sanità, 00161 Rome, Italy; luca.pasquini@iss.it; 4Department of Cardiovascular, Endocrine-Metabolic Diseases and Ageing, Istituto Superiore di Sanità, 00161 Rome, Italy; ornella.morsilli@iss.it

**Keywords:** SARS-CoV-2, coagulation disorders, megakaryocytes, CD147, MCT4

## Abstract

Coagulation disorders are described in COVID-19 and long COVID patients. In particular, SARS-CoV-2 infection in megakaryocytes, which are precursors of platelets involved in thrombotic events in COVID-19, long COVID and, in rare cases, in vaccinated individuals, requires further investigation, particularly with the emergence of new SARS-CoV-2 variants. CD147, involved in the regulation of inflammation and required to fight virus infection, can facilitate SARS-CoV-2 entry into megakaryocytes. MCT4, a co-binding protein of CD147 and a key player in the glycolytic metabolism, could also play a role in SARS-CoV-2 infection. Here, we investigated the susceptibility of megakaryocytes to SARS-CoV-2 infection via CD147 and MCT4. We performed infection of Dami cells and human CD34^+^ hematopoietic progenitor cells induced to megakaryocytic differentiation with SARS-CoV-2 pseudovirus in the presence of AC-73 and syrosingopine, respective inhibitors of CD147 and MCT4 and inducers of autophagy, a process essential in megakaryocyte differentiation. Both AC-73 and syrosingopine enhance autophagy during differentiation but only AC-73 enhances megakaryocytic maturation. Importantly, we found that AC-73 or syrosingopine significantly inhibits SARS-CoV-2 infection of megakaryocytes. Altogether, our data indicate AC-73 and syrosingopine as inhibitors of SARS-CoV-2 infection via CD147 and MCT4 that can be used to prevent SARS-CoV-2 binding and entry into megakaryocytes, which are precursors of platelets involved in COVID-19-associated coagulopathy.

## 1. Introduction

Severe acute respiratory syndrome coronavirus 2 (SARS-CoV-2), responsible for the disease named coronavirus diseases 2019 (COVID-19), often causes airway and pulmonary symptoms [1,2]. In severe cases, COVID-19 patients develop a dysregulated release of cytokines, also called a “cytokine storm”, which produces an excessive inflammatory and immune response, especially in the lungs, leading to acute respiratory distress (ARDS), pulmonary edema, multi-organ failure, and death [3,4]. Although COVID-19 is primarily considered a respiratory disease, many COVID-19 patients experience extra-respiratory symptoms of various severity, including cardiovascular complications and coagulation disorders [5,6]. During the first two years of the COVID-19 pandemic, COVID-19 post-mortem studies have revealed evidence of the direct viral infection of endothelial cells with the accumulation of inflammatory cells and the presence of platelet-rich thrombi in the pulmonary, hepatic, renal, and cardiac microvasculature, while megakaryocytes were in higher than usual numbers in the lungs and heart [5,6,7,8]. Numerous studies have also demonstrated the significant impact of SARS-CoV-2 infection on the hematopoietic system; it was described to induce lymphopenia, neutrophilia, thrombocytopenia, and stress erythropoiesis, important features of COVID-19 disease [9,10,11]. Coagulation disorders, such as hypercoagulation and thrombosis, thrombocytopenia, have been associated with poor outcome and mortality related to severe illness and acute respiratory distress development in COVID-19 patients [5,6,7,8,9]. Thrombocytopenia and/or thrombosis, although rare, have also been reported with the COVID-19 vaccination [12]. Approved vaccines are effective in preventing COVID-19 and severe disease, but the direct contribution of the SARS-CoV-2 spike protein, also used in mRNA vaccines, to platelet defection remains under surveillance and investigation [13,14]. Moreover, whether SARS-CoV-2 can infect megakaryocytes, the precursors of platelets that are involved in thrombotic events in COVID-19 patients and in patients with long post-acute sequelae of COVID-19, referred to as long COVID, needs to be elucidated, especially with the emergence of new SARS-CoV-2 variants [11,12,15]. Furthermore, in addition to the infectious and inflammatory processes that control hematopoiesis in COVID-19, whether SARS-CoV-2 infection and/or COVID-19 vaccination have a direct impact on the maintenance/number and functionality of multipotent hematopoietic stem progenitor cells (HSPCs) in bone marrow and downstream lineage-committed-hematopoietic progenitors (HPCs), then released upon maturation into the blood circulation [10,11,16], is still a matter of debate. A recent study has found a significant decrease in HSPCs in the umbilical cord blood (UCB) obtained from neonates delivered from pregnant women previously infected by SARS-CoV-2 and/or vaccinated, raising concern about the maintenance and proper functions of these HSPCs in the UCB and in the fetal and neonatal hematopoiesis [16].

More recently, SARS-CoV-2 has been shown to interact with platelets and megakaryocytes via an angiotensin-converting enzyme 2 (ACE2)-independent mechanism and to activate platelets through the multifunctional transmembrane glycoprotein CD147 [17,18], found expressed on platelets from healthy donors [18]. The cell surface expression of ACE2, first considered as the main access for SARS-CoV-2 entry into human cells [19,20], affects SARS-CoV-2 spike protein binding and viral infectivity [21], which can also require host cell surface cofactors, such as the host cell transmembrane protease serine 2 (TMPRSS2) and neuropilin-1 [22,23], or some other binding sites, such as CD147 [24,25], to facilitate SARS-CoV-2 entry into target cells expressing very low or undetectable level of host cell surface ACE2 protein [20,21].

CD147 has been previously identified to contribute to Plasmodium invasion and virus infection, including SARS-CoV-2 infection in COVID-19 and associated comorbidities [17,18,24,25,26,27]. CD147 is a potent inducer of extracellular matrix metalloproteinase (EMMPRIN), involved in the regulation of extracellular matrix (ECM) remodeling during physiological and pathological processes such as wound healing, fibrotic and inflammatory-related diseases, and cancer [24,28,29,30,31,32]. CD147 is expressed in numerous cell types, including hematopoietic cells, and exerts pleiotropic functions by interacting with various binding partners, such as the cyclophilin A also involved in COVID-19 [24,28,29,30,31,32] and the monocarboxylate transporter (MCT4), a potential candidate for antiviral drug target to inhibit the replication of SARS-CoV-2 into host cells [33,34]. MCT4 is a high-affinity lactate transporter involved in the maintenance of intracellular pH homeostasis by exporting intracellular lactate when its concentration is high, such as during enhanced aerobic glycolysis that, in case of altered glucose metabolism, may support virus replication in host cells [33,34].

In our study, we have investigated the susceptibility of megakaryocytes (MKs) to SARS-CoV-2 infection via CD147 and MCT4, in vitro. We first analyzed CD147, MCT4 and ACE2 expression during phorbol 12-myristate 13-acetate (PMA)-induced megakaryocytic (MK) differentiation of Dami cells, used as an in vitro model of megakaryopoiesis [35,36,37], and during MK differentiation of CD34^+^ hematopoietic progenitor cells (HPCs) purified from human cord blood (CB) and peripheral blood (PB), as previously described [38]. We found that CD147 and MCT4 are expressed during MK differentiation of PMA-treated Dami cells and CD34^+^ HPCs purified from CB, while, in our set up, ACE2 is undetectable in these cells. However, we also found low expression of ACE2, both at the mRNA and protein level, during MK differentiation of CD34^+^ HPCs purified from PB. Therefore, we performed SARS-CoV-2 pseudovirus infection of PMA-treated Dami cells and MK-differentiating HPCs (MK-HPCs) generated from both human CB and PB CD34^+^ [37,38], in the presence, or absence, of AC-73 or syrosingopine (SYRO), which are specific inhibitors of CD147 and MCT4, respectively [39,40]. Notably, AC-73 and SYRO are inducers of autophagy, a metabolic process essential for megakaryopoiesis [29,41,42,43]. We found that both AC-73 and SYRO enhance the autophagy triggered by PMA during MK differentiation of Dami cells and required for CD34^+^ HPCs survival and differentiation into MKs [42]. However, only AC-73 enhances MK differentiation of PMA-Dami cells and CB- or PB- derived MK-HPCs. AC-73 and SYRO treatment does not have any toxicity or any significant effect on in vitro cell growth or death of MKs. Importantly, we found that AC-73 or SYRO pre-treatment of PMA-Dami cells and MK-HPCs significantly inhibits SARS-CoV-2 pseudovirus infection in these cells.

Altogether, our data indicate AC-73 and SYRO as potential inhibitors of SARS-CoV-2 spike protein binding and entry, via CD147 and MCT4 respectively, into MKs co-expressing CD147 and MCT4, in contrast to ACE2 undetectable or very low expression in MKs derived from CD34^+^ HPCs, and precursors of platelets involved in the SARS-CoV-2 related coagulopathy in COVID-19 disease.

## 2. Materials and Methods

### 2.1. Cell Cultures

Dami cell line. The human megakaryoblastic Dami cell line was maintained in RPMI supplemented with 10% FCS and treated with 100 nM of PMA (Sigma-Aldrich, St. Louis, MO, USA) to inhibit cell growth and induce MK differentiation and maturation, as previously described [35,36,37].

Adult Peripheral Blood and Cord Blood Human Progenitor Cells Purification and MK Culture. Adult PB mononuclear cells were obtained from buffy coats removed from blood donation from normal healthy subjects. CB was obtained after informed consent from healthy donors, according to institutional guidelines and approval by local ethical committees of ISS (no: #171639). Human CD34^+^ cells were purified from CB and PB by positive selection using the midi-MACS immunomagnetic separation system (Miltenyi Biotec, Bergisch Gladbach, Germany) according to the manufacturer’s instructions. The purity of CD34^+^ cells, assessed by flow cytometry using a monoclonal PE-conjugated anti-CD34 antibody was routinely over 95% (range between 92% and 98%) and CD34^+^ HPCs were cultured in serum-free medium in the presence of 100 ng/mL thrombopoietin (TPO) and various cytokines and factors to induce selective megakaryocytic cell differentiation and maturation, giving rise to an almost pure megakaryocytic population, as previously described in [38]. The differentiation stages were characterized during the whole culture by morphologic and phenotype analysis of early and late megakaryocytic markers, CD41 and CD61, CD62P.

Cell morphology was examined after May–Grunwald–Giemsa staining. Morphological analysis was performed to assess both cell maturation and the number of nuclear lobes per cell (600× magnification under a microscope, Eclipse 1000, Nikon, Tokyo, Japan, equipped with a digital camera).

### 2.2. Flow Cytometry Analysis of Cell Surface and Intracellular Antigens

Cytometry analysis was run on a FACSCanto Flow cytometer (Becton Dickinson, Bedford, MA, USA) to analyze the expression of membrane differentiation markers CD41, CD61 and CD62P, using FITC-conjugated anti-mouse CD41, PE-conjugated anti-mouse CD61 and CD62P (AC 1.2) (BD Pharmingen, San Diego, CA, USA) [37,38]. Flow cytometric analysis of CD62P expression was also performed by using CD62P antibody (BD Pharmingen, cat. 31795X) after fixation and permeabilization of the cells with Cytofix/Cytoperm solution, according to the manufacturer’s procedures (BD Pharmingen, cat. 2090KZ). Analysis of CD147 and ACE2 cell surface expression was performed using a FITC-anti-CD147 antibody (BD Pharmingen, cat. 555962) [29] and an APC-anti-Human ACE-2 antibody, respectively (R&D Systems, Minneapolis, MN, USA, cat. FAB933A).

### 2.3. Cell Growth, Viability and Apoptosis Analysis, Cell Cycle Profile, DNA-Ploidy

Cell growth was analyzed by cell counting using Trypan blue. Apoptosis was analyzed by using annexin V-FITC and propidium iodide (PI) apoptosis kits to detect both early and late apoptosis, according to the manufacturer’s instruction (BD Pharmingen). Cell Viability assays were performed with a CellTiter-Glo Luminescent Assay based on quantitation of ATP, an indicator of metabolically active cells (Promega, cat. G7571), to analyze the effect of AC-73 and SYRO, used at different concentrations, on the viability of: Dami cells (AC-73 and SYRO cells), PMA-treated Dami cells (AC-73+PMA, SYRO+PMA cells) and MK-differentiating HPC from CB (AC-73 and SYRO cells); the concentration of drug which exhibited 50% cell viability (IC50) was determined. Luminescence was detected with the EnVision Multimode Plate Reader (Perkin Elmer, Waltham, MA, USA).

Flow cytometry analysis of cell cycle and DNA-ploidy was studied by using the BD Cycletest Plus DNA kit (BD, cat. 340242) to analyze the effects of PMA, AC-73+PMA, or SYRO+PMA treatment on the cell cycle and DNA-ploidy of treated-Dami cells, as compared to untreated Dami cells. All procedures were conducted based on the kit protocol. The method involves dissolving the cell membrane lipids with a nonionic detergent, digesting cellular RNA with enzyme, and stabilizing the nuclear chromatin. Then, PI binds to DNA of the isolated nuclei and the flow cytometer analyzes the light emitted by stained cells. Flow cytometer analysis was performed using a FACSCanto flow cytometer (Becton Dickinson, Bedford, MA, USA).

### 2.4. AC-73 and Syrosingopine Treatment and Sensitivity

AC-73 (3-{2-[([1,1-biphenyl]-4-ylmethyl) amino]-1-hydroxyethyl}phenol) (AN-465/42834501, Specs, The Netherlands) and syrosingopine (SYRO) (Sigma-Aldrich, St. Louis, MO, USA) were dissolved in DMSO and diluted in RPMI, with a final DMSO concentration of no more than 0.2% for all in vitro studies [29,41]. Dose–response and time course analysis were performed in Dami cells and CD34^+^ HPCs, using AC-73 or SYRO at (i) 2.5, 5.0, 10, and 15 µM from 1 to 6 days of Dami cell treatment; (ii) 1.0, 2.5, and 5 µM during MK differentiation of CD34^+^ HPCs; data obtained were compared with 0.2% DMSO-treated cells, indicated as control cells. AC-73 and SYRO treatment started on day 3 of megakaryocytic differentiation of HPCs to avoid blocking cellular proliferation. AC-73 and SYRO were added in cultures every 2 days to maintain their activity.

### 2.5. Quantitative Real-Time RT-PCR Analysis

Total RNAs were extracted using TRIzol reagent and reverse-transcribed, as described [29,41]. Quantitative real-time RT-PCR analysis (qRT-PCR) was performed and normalized with the internal control β-actin (ACTB) using commercial ready-to-use primers/probe mixes (Assays on Demand Products, Applied Biosystems): CD147 (assay Hs 00936295_m1), MCT4 (assay Hs00358829_m1), ACTB (assay Hs 9999903_m1) and ACE2 (assays Hs 01085331_m1 and Hs 01085333_m1) [29,41]. The ABI PRISM 7700 DNA Sequence Detection System (Applied Biosystems, Foster City, CA, USA) was used [29,41].

### 2.6. Western Blot Analysis

Aliquots of 25 and 40 µg of total protein extract were prepared and resolved on 10% and 4–15% mini-Protean TGX precast gels (Bio-Rad, Hercules, CA, USA) for standard denaturing electrophoresis, and transferred to nitrocellulose filters using Transblot-Turbo transfer system, according to manufacturer’s instructions (Bio-Rad). Membranes were treated and incubated with specific antibody, as previously described [29]. Bound antibodies were visualized using the enhanced chemiluminescence technique (ECL) according to the manufacturer’s instructions (WesternBright ECL-spray, Advansta Inc., San Jose, CA, USA). Antibodies used were: JAK2 monoclonal antibody (Jak2 (D2E12)XP #3230 Cell Signaling Technology, Danvers, MA, USA); MPL polyclonal antibody (TpoR/MPL (AF1016) R&D, Minneapolis, MN, USA); phospho-STAT5 polyclonal antibody (phospho-Stat5 (Tyr694) #9351 Cell Signaling Technology, Danvers, MA, USA); phospho-STAT3 polyclonal antibody (phospho-Stat3 (Ser727) #9134 Cell Signaling Technology, Danvers, MA, USA); phospho-ERK1/2 polyclonal antibody (phospho-p44/42 MAPK (Erk1/2) (Thr202/Tyr204) #9101 Cell Signaling Technology, Danvers, MA, USA); total ERK1/2 polyclonal antibody (p44/42 MAP Kinase antibody #9102 Cell Signaling Technology, Danvers, MA, USA), CD147 monoclonal antibody (EMMPRIN/CD147 (B5) sc-46700, Santa Cruz Biotechnology, CA, USA); MCT4 monoclonal antibody (MCT4 (F-10) sc-376101, Santa Cruz, CA, USA). ACE2 polyclonal antibody (# ab15348, Abcam) for Western blot analysis. Anti-ACE2 antibodies tested: anti-ACE2 (#4355, 1:1000) from Cell Signaling Technology (Danvers); anti-ACE2 (#FAB933A, 1:200) from R&D Systems (Minneapolis); anti-ACE2 (# sc-73668, 1:500 and # E-11, sc-390851, 1:500) from Santa Cruz Biotechnology; anti-ACE2 (# 272690, 1:500) from Abcam. LC3B polyclonal antibody (NB600-1384, Novus Biologicals, Novus, I) for autophagy analysis. Luciferase monoclonal antibody (Luciferase (C-12) sc-74548, Santa Cruz, CA, USA). GAPDH polyclonal antibody (Sigma-Aldrich) was used as an internal control for the loaded amounts of total proteins. Bands were quantified by densitometric analysis using the FluorChem E system, software version 4.1.1. The quantification of each band was normalized using the signal of GAPDH as the loading control.

### 2.7. Autophagy Detection

Western blot analysis of the autophagy-related marker LC3 and its conversion from LC3-I to LC3-II form in AC-73- or SYRO-treated cells, as compared to untreated cells of control. Detection and quantification of LC3-I and LC3-II protein expression levels were performed, as described in [29,31].

### 2.8. In Vitro SARS-CoV-2 Infection

In vitro infection of untreated cell cultures or cultures treated for 2 days with AC-73 or SYRO, was performed by using SARS-CoV-2 pseudoviral particles spike, replication-deficient MLV pseudotyped particles with the SARS-CoV-2 spike protein carrying the D614 genotype (MBS434275, MyBiosource, San Diego, CA, USA), to investigate potential inhibitors to block SARS-CoV-2 entry, as described in [44] and according to the manufacturer’s instructions and available protocol (MyBiosource), in a biosafety level 2 laboratory. Then, SARS-Co-V-2 spike protein mediated cell entry was measured via luciferase reporter activity, by using SARS-CoV-2 pseudovirus entry assays (ONE-Glo EX Luciferase Assay System, Promega, Madison, WI, USA), according to the manufacturer’s instructions.

Briefly, for each culture condition, 40,000 cells precipitated by centrifugation were re-suspended in 100 µL of medium containing SARS-CoV-2 pseudoviral particles and seeded in 96-well plates (40,000 cells/100 µL viral particles/well; performed in triplicate) for 3 h of incubation at 37 °C for. Then, 100 µL complete medium (RPMI + 10% FCS for Dami and K562 cells; complete megakaryocytic medium for HPCs [38]) was added in each well and incubated for another 48 h at 37 °C. Infected cells were harvested from each condition of culture (AC-73-treated, SYRO-treated, and untreated cells), precipitated by centrifugation, and re-suspended in 100 µL of buffer from ONE-Glo EX Luciferase Assay, according manufacturer’s instructions (Promega), to measure the luminescence signals using a Victor X Light Luminescence Plate Reader (Perkin Elmer). The luminescence signal detected in each well, reported as relative light units (RLUs), indicates the luciferase reporter activity of the infected cells.

### 2.9. Statistical Analysis

Data are presented as mean ± standard error of the mean (SEM). All data were analyzed using GraphPad Prism 8.0 statistical software. The Shapiro–Wilk test was used to verify normality. Analysis of variance (ANOVA) was used to compare the differences among various groups and Student’s *t*-test was employed to compare the difference between two groups.

Statistical significance was set at *p* < 0.05.

## 3. Results

### 3.1. CD147 and MCT4 Expression during PMA-Induced Megakaryocytic Differentiation of Dami Cells, ACE-2-Deficient Cells

To investigate whether CD147 and MCT4 may facilitate SARS-CoV-2 entry into megakaryocytes (MKs), we first analyzed ACE-2, CD147 and MCT4 expression in an in vitro cellular model of megakaryocytopoiesis, the human megakaryocytic Dami cell line treated with PMA [35]. PMA used at 100 nM inhibits cell growth (Figure 1A) and induces megakaryocytic differentiation of Dami cells, as shown by increased expression of CD41, CD61, and CD62P MK surface antigens (Figure 1B), cell cycle, and DNA-ploidy analysis on day 6 in PMA-treated cells (Figure 1C), as measured by flow cytometry and confirmed by morphology analysis (Figure 1D). On day 6, about 25–30% PMA-treated Dami cells are in G2-M and 10% reach the 4 > 8 N stage via endomitosis (Figure 1C). In addition, we found an increased expression of JAK2 and MPL, the phosphorylation and activation of STAT5 (p-STAT 5), STAT3 (p-STAT 3) and ERK1/2 (p-ERK1/2) (Appendix A), as previously described in Dami cells treated with 100 nM PMA [37]. We also observed the activation of autophagy that accompanies the megakaryopoiesis [42,43] and MK differentiation of Dami cells, as shown by the upregulation of LC3-II expression during MK differentiation of PMA-treated Dami cells (Figure 1E).

Altogether, our data show that 100 nM PMA induces proliferation arrest and megakaryocytic differentiation of the human Dami cell line, mimicking TPO-driven megakaryopoiesis [35,36].

Then, by analyzing CD147, MCT4, and ACE2 expression during PMA-induced MK differentiation of Dami cells, we found that CD147 protein expression level, high in untreated Dami cells (day 0), decreases but remains quite constant during PMA treatment of these cells, as shown by flow cytometry and western blotting analysis (Figure 2A,B). MCT4 protein level increases during the PMA-induced MK differentiation of Dami cells (Figure 2C). ACE2 protein expression was, in our set up, undetectable in Dami cells and PMA-Dami cells as compared to K562 cells used as a positive control for ACE2 expression by western blotting (Figure 2D), flow cytometry, or using several commercially available anti-human ACE2 antibodies. Besides, we could not detect any ACE-2 mRNA expression by using qRT-PCR analysis, in Dami and PMA-Dami cells, as compared to K562 cells.

Altogether, our data show that PMA-treated Dami cells, MK-like cells expressing both CD147 and MCT4 and lacking detectable ACE-2, are a valuable cellular model to investigate the potential role of CD147 and MCT4 in SARS-CoV-2 entry into megakaryocytes.

### 3.2. AC-73 and SYRO Have Both a Synergic Effect on the Autophagy Triggered by PMA during MK Differentiation of Dami Cells, but Only AC-73 Enhances MK Differentiation and Promotes Endomitosis in PMA-Dami Cells

To investigate whether CD147 and MCT4 play a role in SARS-CoV-2 entry into megakaryocytes (MKs), we first analyzed the impact of CD147 and MCT4 blockade using their specific inhibitors, AC-73 and SYRO, respectively, on the proliferation and PMA-induced MK differentiation of Dami cells.

First, we calculated IC50, the concentration of AC-73 and SYRO that exhibit 50% cell viability for Dami and PMA-Dami cells (Appendix A). We performed time– and dose–response analysis using different concentrations of drugs, i.e., from 5 to 20 µM for AC-73 and SYRO. Then, we determined that 10 µM AC-73 and 2.5 µM SYRO inhibit Dami cell growth (Appendix A) without significant apoptosis (Appendix A) but by inducing autophagy, as previously described [29,41] and as shown by the increase in the LC3-II to LC3-I ratio in AC-73 (10 µM)-treated and SYRO (2.5 µM)-treated Dami cells, as compared to untreated Dami cells (C) (Appendix A). Altogether, our data indicate that 10 µM AC-73 and 2.5 µM SYRO are not cytotoxic to Dami cells. However, the use of AC-73 in combination with SYRO to treat Dami cells was toxic. Then, we administered AC-73 (10 µM) or SYRO (2.5 µM) every 48 h during the PMA-induced MK differentiation of Dami cells. AC-73 and SYRO have no significant effect on PMA-Dami cell proliferation (Figure 3A, AC-73+PMA and SYRO+PMA) as compared to PMA treatment (Figure 3A, PMA). AC-73, when used alone has no significant effect on the MK differentiation and maturation of Dami cells, significantly enhances the MK differentiation of PMA-treated Dami cells, as shown by increased CD41 and CD61 expression in (AC-73+PMA)-Dami cells, as compared to PMA-Dami cells, from day 3 to day 6 (Figure 3B). In addition, AC-73 used in combination with PMA promotes endomitosis in (AC-73+PMA)-treated Dami cells, as shown by the noticeable increase in the percent of cells in the G2/M phase and with polyploidy (8N) (Figure 3C), and by morphology analysis (Figure 3D) on day 6 of culture, as compared to PMA-Dami cells and control Dami cells (C). Then, we found that AC-73 has a synergic effect on the autophagy triggered by PMA during MK differentiation of Dami cells (Figure 3E). In contrast, SYRO used alone or in combination with PMA has no significant effect on MK differentiation of (SYRO)-Dami cells and (SYRO+PMA)-Dami cells, as compared to untreated (C) Dami cells and to PMA-Dami cells, respectively, from day 3 to day 6 (Figure 3B). We could not detect any noticeable change in the cell cycle and morphology of (SYRO+PMA)-Dami cells, as compared to PMA-Dami cells, and any relevant synergic effect of SYRO on the PMA-induced autophagy in (SYRO+PMA)-Dami cells, as compared to PMA-Dami cells, from day 3 to day 6 (Figure 3E). 

Altogether, our data show that CD147 blockade by AC-73 enhances autophagy and MK differentiation of PMA-treated Dami cells, promoting endomitosis in these cells, while SYRO has no significant effect on the autophagy triggered by PMA and MK differentiation of PMA-Dami cells.

### 3.3. AC-73 and SYRO Are SARS-COV-2 Entry Inhibitors by Blocking CD147 and MCT4 Function in PMA-Dami Cells

To understand whether CD147 and MCT4 are potential targets for the SARS-CoV-2 spike protein binding to megakaryocytic cells, we used a SARS-CoV-2 pseudovirus to perform a series of infection of Dami and PMA-Dami cells, treated with AC-73 (10 µM) or SYRO (2.5 µM), as compared to untreated cells. 

First, we controlled SARS-CoV-2 infection of Dami cells after 48 h of incubation with SARS-CoV-2 pseudovirus particles by western blotting analysis of the luciferase protein expression in these cells, indicative of the spike protein entry into infected (+) Dami cells, as compared to non-infected (−) cells (Figure 4A), as described [44]. Then, we showed that the inhibition of pseudovirus entry into Dami cells is AC-73 dose-dependent, by measuring the spike protein mediated cell entry into AC-73-Dami cells via detection of luciferase reporter activity with a pseudovirus-based luciferase reporter assay (Figure 4B), as described in [44]. The luciferase activity detected in 10 µM AC-73-Dami cells (Figure 4B, RLU AC-73 10 µM: 57%) was significantly lower, as compared to the luciferase activity detected in 5 µM AC-73-Dami cells and untreated (C) Dami cells reported as RLU 100% (Figure 4B: 5 µM and untreated C), indicating that blockade of CD147 by 10 µM AC-73 impairs pseudovirus entry into Dami cells, although not at 100%, indicating potential other spike protein binding sites on these cells. Higher AC-73 dosages (15 and 20 µM) that decrease the viability of AC-73-Dami cells (Appendix A) are not significantly more efficient than 10 µM AC-73 in blocking pseudovirus entry into Dami cells (Figure 4B).

We also controlled the spike protein entry into K562 cells co-expressing ACE2 and CD147 but negative for MCT4 expression [40], treated with AC-73, as compared to untreated K562 cells. AC-73-treated and untreated K562 cells were incubated 48 h with SARS-CoV-2 pseudovirus particles and measured pseudovirus entry into infected K562 cells, via detection of luciferase reporter activity [44]. The luciferase activity detected in AC-73-treated K562 cells (Figure 4C, RLU AC-73: 52%) is significantly lower, as compared to the luciferase activity detected in untreated (C) K562 cells and reported as RLU 100% (Figure 4C, untreated C), indicating that blockade of CD147 by AC-73 impairs pseudovirus entry into K562 cells.

Then, we performed a series of pseudovirus infection of Dami and day 3 PMA-Dami cells also treated for 2 days with AC-73, as compared to untreated cells, and measured luciferase activities to evaluate SARS-CoV-2 entry into these cells (Figure 4C,D).

Luciferase activity detected in AC-73-Dami cells (Figure 4D, RLU AC-73: 69%) and (AC-73+PMA)-Dami cells (Figure 4E, RLU AC-73+PMA: 54%) is significantly lower than luciferase activity detected in the respective controls reported as RLU 100% (Figure 4D, Dami cells: RLU C and Figure 4E, PMA-Dami cells: RLU PMA: 100%). Luciferase activity detected in SYRO-Dami cells (Figure 4F, RLU SYRO: 32%) is also significantly lower than luciferase activity detected in untreated (C) Dami cells (Figure 4F, RLU C: 100%). 

Altogether, our data indicate that Dami cells are susceptible to infection with SARS-CoV-2 despite the undetectable level of ACE2 in these cells, via CD147 and MCT4. AC-73 and SYRO, respective inhibitors of CD147 and MCT4, are potential SARS-COV-2 entry inhibitors in PMA-Dami cells, megakaryocytic-like cells.

### 3.4. AC-73 and SYRO Are Inhibitors of SARS-COV-2 Entry into Megakaryocytes by Blocking Their Respective Targets CD147 and MCT4

We further investigated whether CD147 and MCT4 are SARS-CoV-2 spike protein binding site on MKs, using CD34^+^ HPCs purified from human CB and adult PB, induced to unilineage MK differentiation in the presence or absence (C) of AC-73 or SYRO, to perform SARS-CoV-2 pseudovirus infection.

First, we used CD34^+^ HPCs purified from human CB that we grew in liquid suspension culture in presence of TPO (100 ng/mL) and various cytokines for a gradual wave of differentiation and maturation along the megakaryocytic lineage characterized by the increased surface expression of megakaryocytic markers CD41, CD61 and CD62P, in a time-dependent manner (Figure 5D,E) [38].

We controlled ACE2 and CD147 expression during MK differentiation of CD34^+^ HPCs. CD147 mRNA expression, high in CD34^+^ HPCs, decreases during the first week of proliferation then remains quite constant in MK-HPCs during the later phase of megakaryocytic differentiation and maturation (Figure 5A), in line with CD147 protein expression analysis by western blotting (Figure 5B). MCT4 mRNA expression increases during MK differentiation of HPCs derived from CB (Figure 5A). ACE2, undetectable by real-time PCR analysis of ACE2 mRNA expression, is also undetectable at the protein level using flow cytometry and western blotting analysis (Figure 5B).

We analyzed the cytotoxic effects of AC-73 and SYRO during MK differentiation of CD34^+^ HPCs derived from CB (Appendix A). Dose–response analysis was performed to analyze the proliferation (Appendix A), evaluate IC50 for both molecules on day 6 of MKs cultures (Appendix A) and apoptosis (Appendix A). Our data show that both AC-73 and SYRO decrease MKs cell growth and viability in a dose-dependent manner. A significant effect on apoptosis was detected by 5 µM for both AC-73 and SYRO, either on day 6 or day 10 of Mks cultures (Appendix A). We then performed in vitro infection of MKs collected on day 10 from cultures treated with AC-73 and SYRO used at 1, 2.5, 5 µM, and incubated 48 h with SARS-CoV-2 pseudovirus particles. All infected cells, day 10 MKs- +AC-73, +SYRO, and day 10 untreated Mks (C), were harvested and lysed to measure the luciferase activity (Appendix A). For both AC-73 and SYRO, used at 2.5 and 5 µM, luciferase activities detected in day 10 treated-MKs are significantly lower than luciferase activity detected in 1 µM AC-73- or SYRO-treated MKs and in untreated control MKs reported as RLU 100% (Appendix A, RLU AC-73 2.5 µM: 60% and AC-73 5 µM: 55%; RLU SYRO 2.5 µM: 55%; SYRO 5 µM: 34%). These data indicate that the inhibition of SARS-CoV-2 entry into MKs is AC-73 and SYRO dose-dependent. However, it is important to consider the significant apoptosis found in MKs treated with 5 µM AC-73 or SYRO (Appendix A).

Then, we used a 2.5 µM dose of AC-73 and SYRO to treat CD34^+^ HPCs, which significantly decreased cell growth of MK-HPCs (+AC-73 and + SYRO, Figure 5C), as compared to untreated (C) MKs, without significant apoptosis. However, only 2.5 µM AC-73 treatment significantly enhances MK differentiation, as compared to 2.5 µM SYRO treatment of MKs and to untreated (C) MKs, as shown by the increased expression of megakaryocytic markers CD41, CD61 (Figure 5D), and CD62P from day 6 to day 13 (Figure 5E) and confirmed by morphology analysis on day 13 (Figure 5F). As observed in PMA-Dami cells (Figure 3E), AC-73 enhances the autophagy required for the survival of CD34^+^ HPCs and their normal differentiation into MKs [42], as shown by increased LC3-II to LC3-I ratio in AC-73 treated (+) MKs on day 7 and day 14 (d7+, d14+, Figure 5G), as compared to untreated (−) day 7 and day 14 MKs and control CD34^+^ cells (Figure 5G).

Then, to perform in vitro infection of MKs collected on day 10 of culture, pre-treated with AC-73, SYRO (2.5 µM) or no pretreatment, we incubated these cells for 48 h with SARS-CoV-2 pseudovirus particles. All infected cells, day 10 Mks+AC-73-, MKs+SYRO- and untreated MKs- HPCs, were harvested and lysed to measure the luciferase activity. Luciferase activity detected in day 10 Mks+AC-73 and day 10 Mks+SYRO is significantly lower than luciferase activity detected in untreated day 10 MKs and reported as RLU 100% (Figure 5H, RLU d10 MK+AC-73: 60%; RLU d10 MK+SYRO: 58%), indicating that MKs are susceptible to SARS-CoV-2 entry via CD147 or MCT4, which can be blocked by AC-73 or SYRO, respectively.

Then, we performed SARS-CoV-2 pseudovirus infection experiments using CD34^+^ HPCs purified from adult PB and induced to unilineage MK differentiation [38], in presence of 2.5 µM dose of AC-73 or SYRO, as compared to untreated MKs cultures, to investigate whether CD147 and MCT4 are potential targets for SARS-CoV-2 entry into host cells, such as MKs in adults, and then generating platelets in blood stream. 

First, we controlled ACE-2, CD147, and MCT4 expression during MK differentiation of CD34^+^ HPCs. CD147 expression level decreases but remains always detectable during MK differentiation of CD34^+^ HPCs purified from PB (Figure 6A), in line with our data obtained for MKs derived from CD34^+^ HPCs purified from CB (Figure 5A,B). MCT4 mRNA and protein expression increases during MK differentiation (Figure 6A,B). Surprisingly, we detected a very low level of ACE2 mRNA expression by real-time PCR analysis, which increases at a later stage (day 13) of MK differentiation (Figure 6A) and was also detectable, even though at low level, on day 13 by western blotting analysis of ACE2 protein expression (Figure 6B).

Then, we analyzed the impact of AC-73 and SYRO treatment during MK differentiation of PB CD34^+^ HPCs and found that neither AC-73 (2.5 µM) nor SYRO (2.5 µM) have a significant effect on cell growth during MK differentiation of CD34^+^ HPCs (Figure 6C). As previously found in PMA-Dami cells (Figure 3E) and in MKs from CB (Figure 5G), both AC-73 and SYRO enhance the autophagy required for PB CD34^+^ HPCs differentiation into MKs (Figure 6D), while AC-73 treatment, but not SYRO, enhances MK differentiation of CD34^+^ HPCs from PB, as shown by the significant increase in CD62P membrane expression in MKs+AC-73 (Figure 6E, +AC-73), as compared to SYRO treatment of MKs (Figure 6E, +SYRO) and untreated (C) Mks (Figure 6C,E), from day 5 to day 13. 

Then, to perform in vitro infection of MKs collected on day 10 of cultures, treated with AC-73 (2.5 µM) and SYRO (2.5 µM) and untreated, we incubated these cells for 48 h with SARS-CoV-2 pseudovirus particles. All infected cells, day 10 MKs+AC-73, day 10 MKs+SYRO and untreated day 10 MKs, were harvested and lysed to measure the luciferase activity. Luciferase activities detected in MKs+AC-73 and MKs+SYRO are significantly lower than luciferase activity detected in untreated day 10 MKs and reported as RLU 100% (Figure 6F, RLU d10 MK+AC-73: 44%; RLU d10 MK+SYRO: 34%), indicating that Mks are susceptible to SARS-CoV-2 entry via CD147 or via MCT4, which can be blocked, respectively, by AC-73 and SYRO.

Altogether, our data show that AC-73 and SYRO are inhibitors of SARS-CoV-2 entry, respectively, via CD147 and MCT4 targeting, as an alternative to ACE2, into human CB- and adult PB-derived megakaryocytic cells.

## 4. Discussion

After 3 years of the COVID-19 pandemic in Europe and Central Asia, the World Health Organization has recently declared the end of the COVID-19 global public health emergency. We are now entering a new phase to strengthen surveillance, particularly for the emergence of new SARS-CoV-2 variants or other viruses, to develop next generation of vaccines and therapies, and to challenge long COVID, a multisystemic condition following SARS-CoV-2 infection often comprising severe symptoms [15,45]. The pathogenetic mechanisms leading to severe COVID-19 disease involve endothelial dysfunction, hyperactivation of the immune system and immune-thrombosis; these all require early intervention to prevent the high risk of venous thromboembolism that may lead to death [4,5,6,7,8,9] or to the long-term residual effects of COVID-19-associated coagulation found in long COVID patients [46]. Recent studies have reported the possibility of viral persistence as a driver of long COVID [43,44]; viral proteins and/or RNA have indeed been found in lymph nodes, hepatic tissue, lung tissue, plasma, stool, and urine of some patients [47,48]. Studies performed by analyzing the coagulation profile of long COVID patients have described a pro-coagulant state with an ongoing process of thrombi formation and/or persistent microthrombosis in 30% of patients after 12–18 months of follow-up [46], underlying the need to further investigate the etiology of COVID-19-associated coagulopathy for the early management of coagulation disorders [49,50]. However, the mechanism promoting platelet activation by SARS-CoV-2 is still not well understood [51]. Nonetheless, the observation that megakaryocytes (including lung MKs) rather than platelets may directly interact with SARS-CoV-2 in the blood, is supported by several studies [52,53]. Importantly, as lung epithelial cells are the primary site of SARS-CoV-2 infection, and therefore are responsible for initiating immune responses to virus infection, an increase in the number of lung and circulating megakaryocytes has been described in COVID-19 patients [54]. In line with previous data [17,18], we observed that SARS-CoV-2 can interact, as an alternative to ACE2, with megakaryocytes via CD147, that we found expressed in human CB and PB CD34^+^ HPCs, and during MK differentiation of these cells, thus indicating megakaryocytes as potential direct target of SARS-CoV-2. CD147, expressed on the cell surface of leukocytes, plays a crucial role in the recruitment of innate or adaptative immune cells required to fight the virus infection and consequent inflammation [24,27,28,29,32]. In the context of thrombo-inflammation, CD147 induces leukocyte chemotaxis and adhesion, as well as platelet activation and the subsequent thrombus formation through the binding of various interaction partners [24,28,32,55]. More recently, MCT4, co-expressed with CD147 on leucocytes and platelets membrane to facilitate lactate efflux from highly glycolytic cells, has been indicated as a potential target for antiviral drugs to prevent SARS-CoV-2 replication in host cells, with viral replication being supported by enhanced aerobic glycolysis in these cells [24,56]. Our data show that MCT4 expression increases during MK differentiation of PMA-Dami cells and CD34^+^ HPCs, possibly indicating an increased glycolytic metabolism in MK-differentiating HPCs. These data are in line with previous studies indicating that glucose metabolism is essential for pro-platelet formation from megakaryocytes, platelet biogenesis, and activation [57,58]. Furthermore, we found that MCT4 is a target for SARS-CoV-2 entry into PMA-Dami cells and MK-differentiating HPCs in vitro. 

Altogether, our data indicate the glycolysis-related MCT4 as a potential new target for SARS-CoV-2 entry into megakaryocytes and platelets in vivo, which may be involved in the hypothetical reservoir for SARS-CoV-2 described in long COVID [15,45].

In our experience, CD147 and MCT4 blockade provided by their respective inhibitors, AC-73 and SYRO, is associated with increased autophagy [29,41], an essential process in megakaryopoiesis, megakaryocyte differentiation, thrombopoiesis, and platelet production [42]. Our study shows that AC-73 and SYRO, by inhibiting CD147 and MCT4 function, respectively, during MK differentiation of PMA-Dami cells and CD34^+^ HPCs, impair SARS-CoV-2 entry into these cells. The treatment of PMA-Dami cells and CD34^+^ HPCs with AC-73 or SYRO, both inducers of autophagy [29,41], has a synergic effect on the autophagy that accompanies MK differentiation of these cells. However, only AC-73 significantly enhances MK differentiation of CD34^+^ HPCs. CD147 blockade by AC-73 that affects CD34^+^ HPCs proliferation, as previously described [29,41], induces MK differentiation, CD147 being highly expressed in these cells and during all steps of differentiation. The late increase in MCT4 expression during MK differentiation could explain the absence of a significant effect on MK maturation during SYRO treatment.

Altogether, our data indicate CD147 and MCT4 as potential binding sites for the SARS-CoV-2 spike protein entry into megakaryocytes and as therapeutic targets for developing specific and effective drugs to block SARS-CoV-2 entry into CD147/MCT4-expressing cells. In particular, our study points to MCT4 as a potential therapeutic target and to SYRO to prevent SARS-CoV-2 entry and replication into MCT4 expressing-MK cells in which the high glycolytic metabolism could sustain viral replication.

Furthermore, our in vitro models of human CD34^+^ HPC-induced MK differentiation represent a tool to study the interactions of SARS-CoV-2 variants with specific hematopoietic target cells and to test new drugs preventing SARS-CoV-2 binding and subsequent infection of these cells.

## Figures and Tables

**Figure 1 viruses-16-00082-f001:**
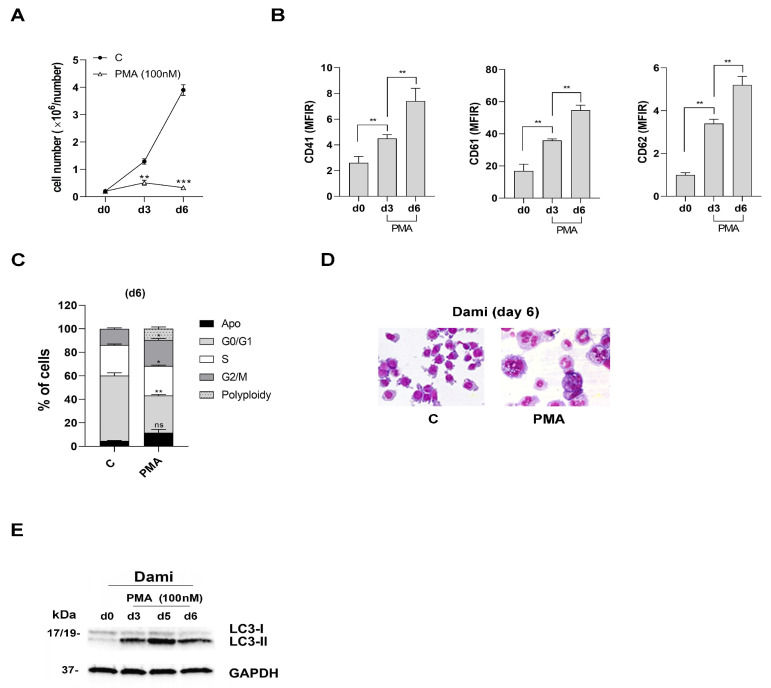
PMA inhibits proliferation and induces MK differentiation and autophagy that accompanies increased ploidy in PMA-Dami cells. (**A**) Cell growth analysis of PMA-Dami cells, in presence of PMA used at 100 nM and added every 2 days in cultures, as compared to untreated Dami cells of control (C). (**B**) Flow cytometry analysis of CD41, CD61 and CD62P increased expression during PMA treatment of Dami cells, from day 0 to day 6 of treatment. (**C**) Flow cytometry analysis of cell cycle and DNA-ploidy status of PMA-Dami cells on day 6, as compared to untreated (C) Dami cells. Percentage of cells at each phase of the cell cycle and DNA-ploidy are shown. (**D**) Morphology analysis of PMA-Dami (PMA) cells on day 6 is shown, as compared to untreated Dami cells (C); (**E**) Western blotting analysis of the autophagy-related protein LC3-II that increases during PMA treatment of Dami cells. GAPDH is shown as an internal control. (**A**–**C**) Mean (± SEM) of three independent experiments is shown. * *p* < 0.05; ** *p* < 0.01; *** *p* < 0.001; ns is for not significant. (**D**,**E**) One representative experiment of three is shown.

**Figure 2 viruses-16-00082-f002:**
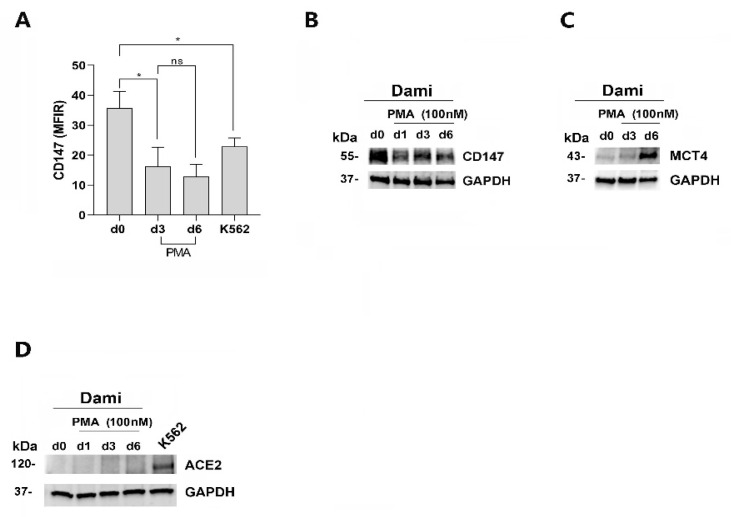
CD147 and MCT4 expression during PMA-induced MK differentiation of Dami cells, negative for ACE-2 expression. (**A**,**B**) CD147 protein expression decreases during the first days (from day d0 to day d3) of PMA treatment of Dami cells, but remains constant and well detectable from day 3 to day 6 of PMA treatment of Dami cells, as shown by flow cytometry analysis (**A**) and Western blot analysis (**B**). (**C**) MCT4 protein expression increases during PMA-induced MK-differentiation of Dami cells, as shown by western blot analysis. (**D**) ACE-2 protein expression is undetectable by western blot analysis, in Dami cells (d0) and during PMA-induced MK differentiation of these cells, as compared to K562, ACE-2 expressing cells. (**A**) Mean ± SEM of three independent experiments is shown. * *p* < 0.05; ns is for not significant. (**B**–**D**) One representative experiment out of three is shown; GAPDH is shown as an internal control. (**D**) K562 cells are a positive control for ACE-2 expression.

**Figure 3 viruses-16-00082-f003:**
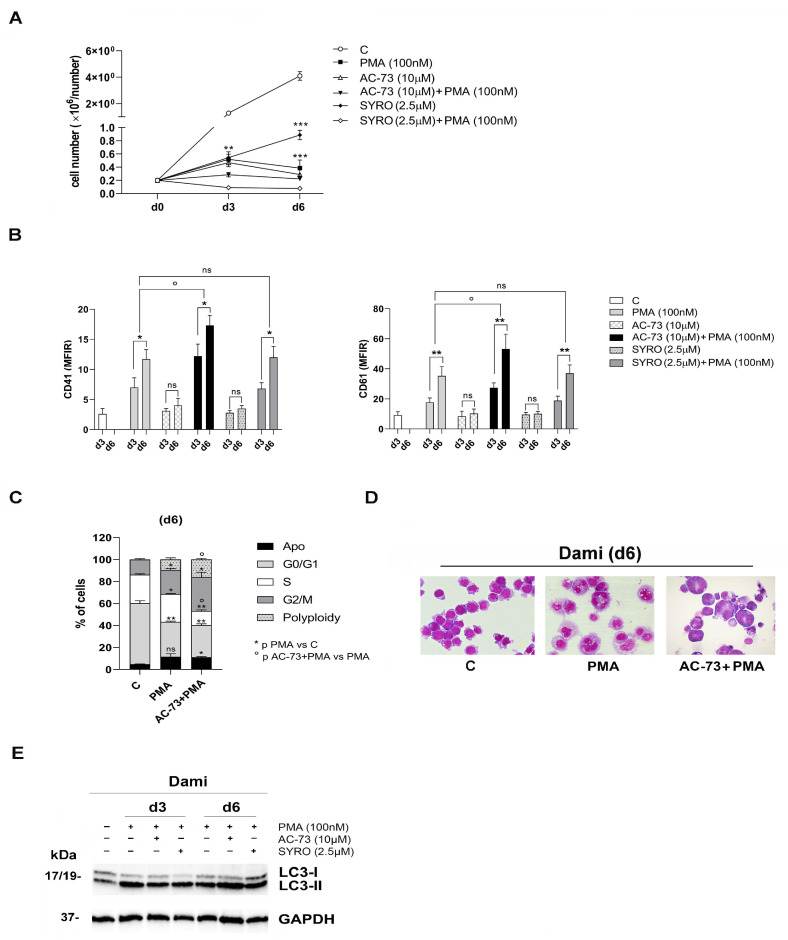
AC-73 and SYRO have a synergic effect on the autophagy triggered by PMA during MK differentiation of Dami cells, but only AC-73 enhances MK differentiation and maturation in PMA-Dami cells. (**A**) Growth curve analysis indicates that (i) used alone, PMA (100 nM), AC-73 (10 µM), or SYRO (2.5 µM) inhibit Dami cell proliferation as compared to untreated (C) Dami cells; (ii) used in combination with PMA treatment, AC-73 (PMA+AC-73) and SYRO (PMA+SYRO) have no significant effect on PMA-Dami cell proliferation, as compared to PMA treatment (PMA). (**B**) Flow cytometry analysis shows a significant increase in CD41 and CD61 expression in co-treated (AC-73+PMA)-Dami cells but not in co-treated (SYRO+PMA)-Dami cells as compared to PMA-Dami cells. (**C**) Flow cytometry analysis of cell cycle and DNA-ploidy status shows a significant increase in the percent of cells in phase G2/M and polyploidy (8N) on day 6 in (AC-73+PMA)-treated Dami cells as compared to PMA-Dami cells, and in PMA-Dami cells as compared to untreated (C) Dami cells. (**D**) Morphological analysis performed on day 6 shows an increase in polyploidy in (AC-73+PMA)-treated Dami cells, as compared to PMA-treated and untreated (C) Dami cells. (**E**) Used in combination with PMA, AC-73 and SYRO enhance the autophagy activated during PMA-induced MK differentiation of Dami cells from day 3 to day 6 of culture as compared to untreated Dami cells, as shown by western blotting analysis of the autophagy-related protein LC3 and its conversion from LC3-I to LC3-II in these cells. (**A**–**C**) Mean ± SEM of three independent experiments is shown. * and ° *p* < 0.05; ** *p* < 0.01; *** *p* < 0.001; ns is for not significant. (**D**) One representative morphological analysis is shown. (**E**) One representative experiment out of three is shown; GAPDH is shown as an internal control.

**Figure 4 viruses-16-00082-f004:**
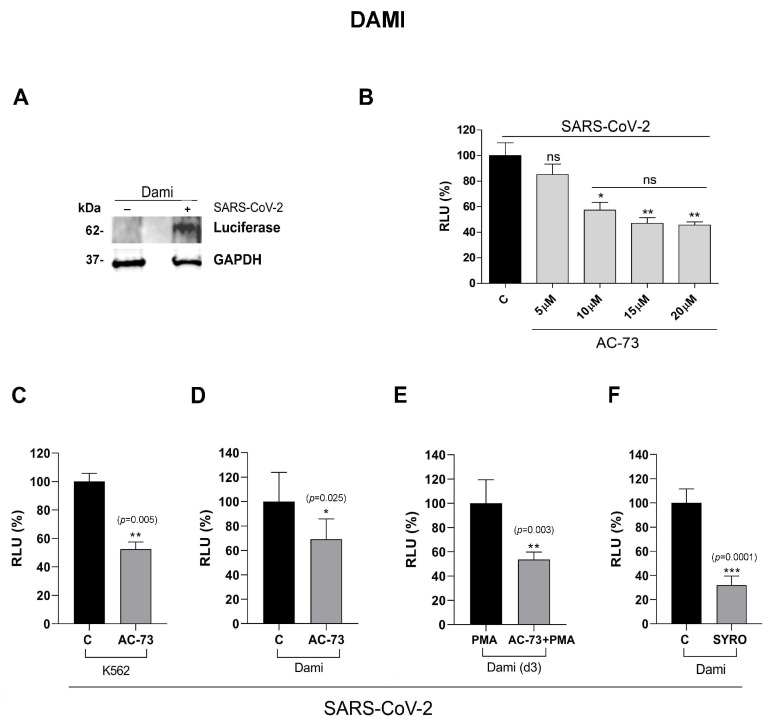
AC-73 and SYRO inhibit SARS-CoV-2 pseudovirus entry into Dami and PMA-Dami cells. (**A**) Western blotting analysis of luciferase protein expression in SARS-CoV-2 infected (+) Dami cells, as compared to non-infected (−) Dami cells, indicating SARS-CoV-2 spike protein entry into infected Dami cells. (**B**) Luciferase activity detected in Dami cells treated with increasing doses (5, 10, 15, and 20 µM) of AC-73, after 2 days of incubation with SARS-CoV-2 pseudovirus particles, is significantly lower in 10, 15, and 20 µM AC-73-treated Dami cells than luciferase activity detected in 5 µM AC-73-treated and untreated (C) Dami cells. (**C**) Luciferase activity detected in AC-73 (10 µM)-treated K562 cells incubated for 2 days with SARS-CoV-2 pseudovirus for infection is significantly lower (RLU AC-73: 52%) than the luciferase activity detected in untreated (C) K562 cells, for which infection is reported as RLU 100%. (**D**,**E**) Luciferase activity detected in AC-73 (10 µM)-treated Dami cells (RLU AC-73: 69%) and (AC-73+PMA)-treated Dami cells (RLU AC-73+PMA: 54%), incubated for 2 days with SARS-CoV-2 pseudovirus particles, is significantly lower than the luciferase activity detected in respective controls, untreated (C) infected-Dami cells (**D**) and infected-PMA-treated Dami cells (**E**), for which infection is reported as RLU 100%. (**F**) Luciferase activity detected in SYRO (2.5 µM)-treated Dami cells (RLU SYRO: 32%), incubated for 2 days with SARS-CoV-2 pseudovirus particles, is significantly lower than the luciferase activity detected in untreated (C) infected-Dami cells (RLU C: 100%). (**A**) One representative experiment out of three is shown; GAPDH is shown as an internal control; molecular weight (kDa) is indicated. (**B**–**F**) Mean ± SEM of three independent experiments is shown. * *p* < 0.05; ** *p*< 0.01; *** *p*< 0.001; ns is for not significant. RLU, relative light unit.

**Figure 5 viruses-16-00082-f005:**
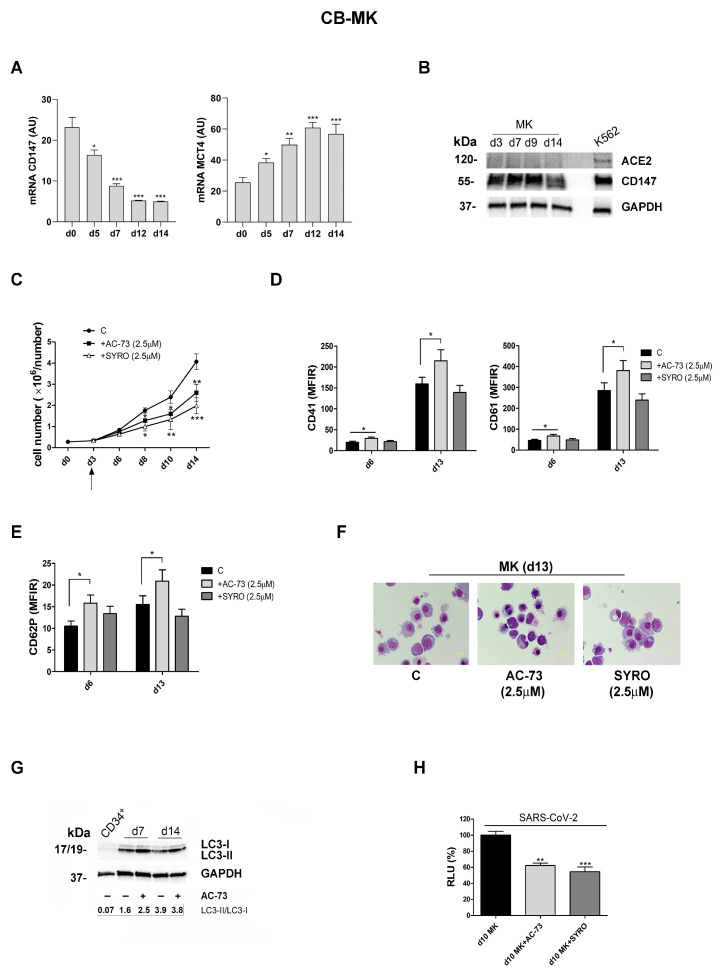
AC-73 treatment enhances autophagy, promotes MK differentiation of CD34^+^ HPCs purified from human CB and inhibits SARS-CoV-2 pseudovirus entry into MK-HPCs. (**A**) CD147 and MCT4 mRNAs are inversely expressed during MK differentiation of CD34^+^ HPCs, as shown by qRT-PCR analysis. (**B**) CD147 protein expression level is high, while ACE-2 protein expression is undetectable in MKs, as shown by western blotting analysis. (**C**) The treatment of 2.5 µM AC-73 and SYRO started at day 3 (arrow) and added every 2 days in MKs cultures significantly impair cell growth of MKs as compared to untreated (C) MKs (**D**–**F**) AC-73, but not SYRO, significantly enhances MK differentiation and maturation, as shown by the major increase of CD41, CD61, and CD62P expression detected on cell surface of MKs+AC-73 on day 6 and day 13, as compared to MKs+SYRO and to untreated (C) MKs, and by (**F**) morphology analysis. (**G**) AC-73 enhances the autophagy required for MK differentiation of HPCs, as shown by western blotting analysis of autophagy-related protein LC3 and its conversion from LC3-I to LC3-II on day 7 and day 14 in AC-73 treated (+) MKs, as compared to untreated (− AC-73) MKs on day 7 and day 14 and to CD34^+^ HPCs. Densitometric data for the LC3B-II to LC3B-I ratio are shown. (**H**) Luciferase activity detected on day 10 of culture in MKs+AC-73 (2.5 µM) (RLU d10 MK+AC-73: 60%) and in MK+SYRO (2.5 µM) (RLU d10 MK+SYRO: 58%), incubated 2 days with SARS-CoV-2 pseudovirus particles, is significantly lower than the luciferase activity detected on day 10 of culture in untreated day 10 MKs (RLU d10 MK: 100%). (**A**,**C**–**E**,**H**) Mean ± SEM of three independent experiments is shown. * *p* < 0.05; ** *p* < 0.01; *** *p* < 0.001. (**B**,**G**) One representative western blot out of three is shown; GAPDH is shown as an internal control. (**F**) One representative morphological analysis is shown.

**Figure 6 viruses-16-00082-f006:**
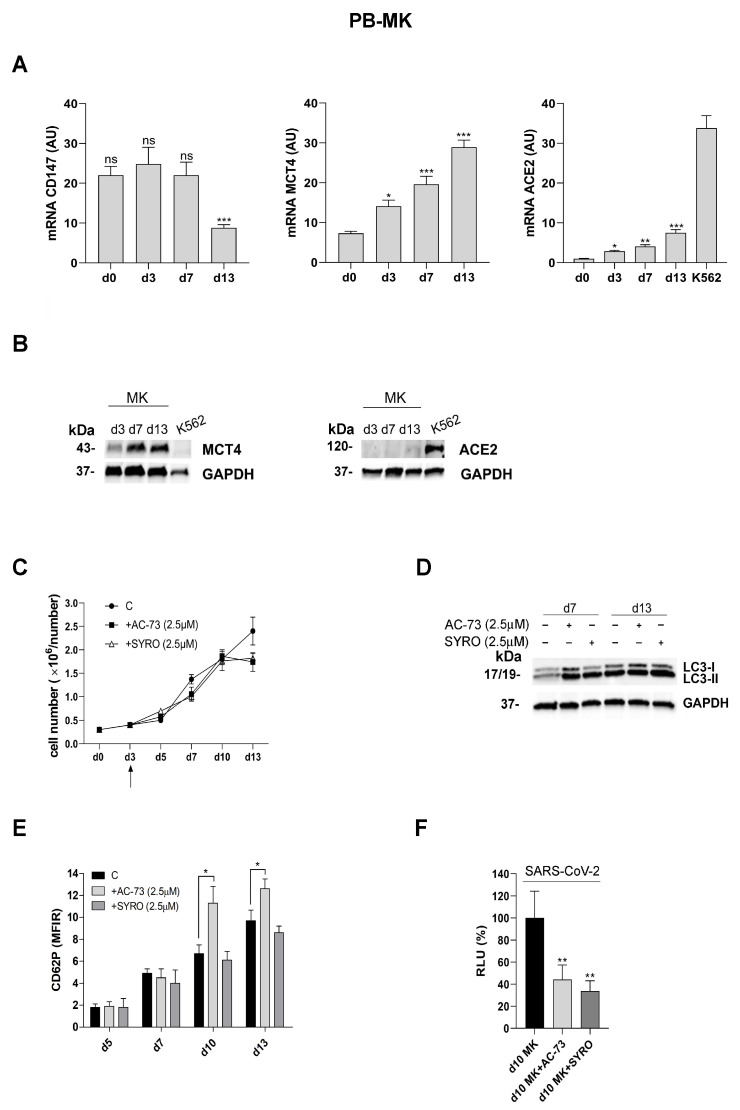
AC-73 and SYRO, which enhance autophagy during MK differentiation of CD34^+^ HPCs purified from human PB, expressing CD147, MCT4, a low level of ACE-2, without significant effect on MK differentiation for SYRO, both inhibit SARS-CoV-2 pseudovirus entry into MK-HPCs from PB. (**A**) CD147 and MCT4 mRNA are inversely expressed, while ACE-2 mRNA expression is also detected during MK differentiation of CD34^+^ HPCs purified from PB, as shown by qRT-PCR analysis. (**B**) MCT4 protein expression increases during MK differentiation of CD34^+^ HPCs from PB, while ACE-2 protein expression is detectable, even though at very low level, on day 13 of MK differentiation, as shown by western blotting analysis. (**C**) The treatment of 2.5 µM AC-73 and SYRO treatment, started on day 3 (arrow) and added every 2 days in megakaryocytic cultures, had no significant effect on cell growth of MKs, as compared to untreated (C) MKs. (**D**) AC-73 and SYRO enhance the autophagy required for MK differentiation of HPCs, as shown by western blotting analysis of autophagy-related protein LC3 and its conversion from LC3-I to LC3-II on day 7 and day 13 in AC-73 or SYRO treated (+) MKs, as compared to untreated (−) day 7 and day 13 MKs and to control CD34^+^ cells. (**E**) AC-73 significantly enhances MK differentiation, as shown by the increased expression of CD62P in MKs from day 5 to day 13, as compared to SYRO, without significant effect on MK differentiation of HPCs (+SYRO) and to untreated (C) MKs. (**F**) Luciferase activities detected on day 10 of culture in AC-73 (2.5 µM) and SYRO (2.5 µM)-treated MKs (RLU d10 MK+AC-73: 44%; RLU d10 MK+SYRO: 34%), incubated 2 days with SARS-CoV-2 pseudovirus particles, are significantly lower than the luciferase activity detected at day 10 in untreated MKs (RLU d10 MKs: 100%). (**A**,**C**,**E**,**F**) Mean ± SEM of three independent experiments is shown. * *p* < 0.05; ** *p* < 0.01; *** *p* < 0.001; ns is for not significant. (**B**,**D**) One representative western blot out of three is shown; GAPDH is shown as an internal control; molecular weight (kDa) is indicated.

## Data Availability

Data available on request.

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
