# Peer review of "AC-73 and Syrosingopine Inhibit SARS-CoV-2 Entry into Megakaryocytes by Targeting CD147 and MCT4"

_viruses, 2024, doi:10.3390/v16010082_

Round 1

Reviewer 1 Report (Previous Reviewer 2)

Comments and Suggestions for Authors

The manuscript is much improved after revision. I believe that the manuscript is ready for publication.

Author Response

Please see attachment: Reviewer 1. docx

Reviewer 2 Report (New Reviewer)

Comments and Suggestions for Authors

The authors have addressed majority of the concerns raised by reviewers. Some concerns that still remain are:

1.     The authors have performed the cell viability assay for AC-73 and SYRO in DAMI and CB-MK cells and have reported the inhibition of pseudotyped SARS-CoV-2 in treated cells. At the tested concentration of 10 µM for AC-73 (Figure 4B and 4D-E), DAMI cells show <50% viability (Figure S2A-B). Similarly, both these tested compounds show 40% toxicity in CB-MK cells at 2.5 µM (Figure S3 C-D), and the same concentration was effective in the reduction of absolute RLU values for pseudotyped SARS-CoV-2-infected cells (Figure S3G). Reporting the inhibition of the virus at this toxic concentration with absolute RLU value as the readout doesn’t make the inhibition convincing.

2.     Elaborate on the rationale behind using K562 cell line to examine the role of CD147 in SARS-CoV-2 entry. This cell line expresses ACE2, which can be a confounding variable in the study. Use of an ACE-2 knockdown or knockout K562 cell line would be rationally more justified.

3.     At some places the manuscript looks uneven in terms of data presentation. Eg.

·       A dose-dependent inhibition of pseudotyped SARS-CoV-2 by AC-73 in DAMI cells is plotted in Figure 4B, abruptly followed by single dose inhibition with AC-73 (PMA + and -) and then single dose inhibition by SYRO (Figure 4F).

·       Both Figure 6F and Figure 3G show the inhibition of pseudotyped SARS-CoV-2 in CB-MK. This data could be plotted in a single graph.

Minor comments:

Introduce the full form of TPO in the text.

Line 66: decreased> decrease

Line 76 impact on>affects

Comments on the Quality of English Language

Please revise the following sentences:

Line 154-157: CD147 cell surface expression (FITC-anti-CD147 antibody from BD Pharmingen) [29] and ACE2 surface expression (APC-anti-Human ACE-2 Polyclonal Antibody from R&D Systems #FAB933A)

Line 252-255: Luciferase activity detected in infected AC-73- or SYRO- treated cells, as compared to infected untreated cells. Firefly luciferase assay Chemiluminescent signals generated by pseudovirus particle entry into K562 cells treated 2 days.

Line 283-285: Altogether, our data show that 100 nM PMA induced proliferation arrest and megakaryocytic differentiation of the human Dami cell line, such mimicking TPO-driven megakaryopoiesis.

Author Response

Please see attachment: Reviewer 2.

This manuscript is a resubmission of an earlier submission. The following is a list of the peer review reports and author responses from that submission.

Round 1

Reviewer 1 Report

Comments and Suggestions for Authors

The aim of Dr Spinello and collaborator’s manuscript was to analyze the effect of two inhibitors to prevent the SARS-CoV-2 infection on megakaryocytes.

Even if the manuscript is well written, several aspects should be addressed to achieve the original goals.

Major concerns

-       Lines 56-59: Do the authors think that the platelets derived from SARS-CoV-2 infected megakaryocytes could be more reactive? This concept could be interesting to demonstrate.

-       Regard to the infection, how viruses can to achieve vascular niche in bone marrow? There is any experimental evidence of this phenomenon?

-        Also if PMA could to induce megakaryocytic differentiation, it is necessary to compare this differentiantion with the specific agonist, thrombopoietin. In particular because PMA could also modify secretory granules distribution.

-       In fact, the authors observed an upregulation of P-selectin (CD62P) on megakaryocytes stimulated with PMA. P-selectin is produced by megakaryocytes and accumulated in internal face of the membrane of alpha granules. In normal conditions, megakaryocytes do not release their granules. In the same line, the CD61 expression on megakaryocytes membrane in intact cells is very low.

-       In methods, the authors described the use of CD34 cells stimulated with TPO but only part of these results are described.

-       The specific effect of AC-73 and SYRO on megakaryocytes should be tested

-       To quantify the effect of the above inhibitors, a specific blocking monoclonal antibody should be tested and compare the results.

-       Statistical analysis: “were performed using paired t-test”. The groups of comparations are more than two thus is necessary to use ANOVA. In both cases it is necessary to confirm the normally distribution of the variables analyzed.

-       The method of flow cytometry analysis of cell cycle and DNA-ploidy is not described.

-       The quality of western blot (Figures 1-5) is very poor. Furthermore, the housekeeping and molecular weight scale should be present in the same gel.

Minor concerns

-       Platelets are not “involved in coagulation disorders” but in haemostasis and thrombosis.

-       Flow cytometry analysis and FACS are not synonymous. Actually, FACs is a brand.

Reviewer 2 Report

Comments and Suggestions for Authors

In the manuscript, Isabella Spinello et al. investigated the susceptibility of megakaryocytes to SARS-CoV- 2 infection. The data showed that AC-73 and Syrosingopine inhibits SARS-CoV-2 binding and entry by targeting CD147 and MCT4. It is interesting that the authors established an in vitro system for SARS-CoV-2 infection. However, the results and conclusions need to be improved. The specific comments are listed below.

1)      The authors need to determine the cytotoxic effect of these two drugs on the cells.

2)      In Figure 4A, 5F, and 6E, the authors tested the effect of AC-73 and Syrosingopine on SARS-CoV-2 infection. The authors need to include multiple concentrations of inhibitors to show if the inhibitory effect of these drugs is dose dependent. Meanwhile, the authors also need to specify the IC50 (50% inhibitory concentration) of these inhibitors for virus in different cell types.

3)      The authors pointed out that AC-73 and Syrosingopine inhibit SARS-CoV-2 entry into megakaryocytes by targeting CD147 and MCT4. However, AC-73 and Syrosingopine not only block CD147 and MCT4, but also induce autophagy. Thus, the conclusion is not convincing. The authors should include the data to show the effect of these inhibitors on the specific step of virus life cycle including virus attachment, internalization as well as replication.

4)      The inhibitory effect of these two drugs is not highly potent varying from 32% to 75%. Please discuss the factors contributing the variations. It would be nice to know the efficiency of SARS-CoV-2 entry into megakaryocytes when CD147 and MCT4 are depleted via knock-down or knock-out.

5)      The authors determined the luciferase activity at different times in the cell cultures using SARS-CoV-2 pseudovirus. The authors need to give explanation about the time point for sampling and show the kinetics of pseudovirus in these cells.

6)      The authors treated the cells with AC-73 and Syrosingopine individually, what data the authors can get when the cells were treated with both AC-73 and Syrosingopine.

7)      The infection of SARS-CoV-2 in the cell culture model in this manuscript is productive or abortive. Please show the data.

Comments on the Quality of English Language

The quality of English is good for me.